# Low-Density Lipoprotein Cholesterol and the Risk of Rheumatoid Arthritis: A Prospective Study in a Chinese Cohort

**DOI:** 10.3390/nu14061240

**Published:** 2022-03-15

**Authors:** Hannah VanEvery, Wenhao Yang, Jinmei Su, Nancy Olsen, Le Bao, Bing Lu, Shouling Wu, Liufu Cui, Xiang Gao

**Affiliations:** 1Department of Nutritional Sciences, The Pennsylvania State University, Pollock Road, University Park, PA 16802, USA; hqv5028@psu.edu; 2Department of Rheumatology and Immunology, Kailuan General Hospital, Xinhua E Avenue, Tangshan 063007, China; ywhywhywh123@163.com (W.Y.); cuiliufu@hotmail.com (L.C.); 3Department of Rheumatology, Peking Union Medical College Hospital, Dongdan N Street, Beijing 100006, China; jxs1246@psu.edu; 4Division of Rheumatology, Department of Medicine, Penn State Milton S. Hershey Medical Center, University Drive, Hershey, PA 17033, USA; nolsen@pennstatehealth.psu.edu; 5Department of Statistics, The Pennsylvania State University, Pollock Road, University Park, PA 16802, USA; lebao@psu.edu; 6Department of Medicine, Harvard Medical School, Shattuck Street, Boston, MA 02115, USA; blu1@bwh.harvard.edu; 7Department of Cardiology, Kailuan General Hospital, Xinhua E Avenue, Tangshan 063007, China; drwusl@163.com; 8Department of Nutrition and Food Hygiene, Fudan University, Dongan Road, Shanghai 200032, China; 9Key Laboratory of Public Health Safety of Ministry of Education, Fudan University, Dongan Road, Shanghai 200032, China

**Keywords:** rheumatoid arthritis, prospective, cohort, cholesterol, epidemiology

## Abstract

Objective: This study aimed to investigate whether low-density lipoprotein cholesterol (LDL-C) concentration was associated with the risk of rheumatoid arthritis (RA) in Chinese adults. Methods: The study included the 97,411 participants in the Kailuan Study without RA, with complete baseline LDL-C data, and who did not use lipid-lowering medications at baseline or during follow-up. We used Cox proportional hazards modeling to calculate the hazard ratio (HR) and 95% confidence interval (95% CI) of RA according to baseline LDL-C tertiles, adjusting for age, sex, body mass index, HDL-C, triglycerides, diabetes, hypertension, alcohol consumption, and smoking. We also calculated the HR and 95% CI of RA using updated LDL-C measurements prior to the end of follow-up, adjusting for covariates. Results: We identified 97 incident RA cases between 2006 and 2018. After adjusting for potential confounders, updated LDL-C concentration—rather than baseline LDL-C—was inversely associated with RA risk. The adjusted HR of RA was 0.64 (95% CI: 0.38, 1.09; *p*-trend = 0.10) comparing the two extreme baseline LDL-C tertiles, and 0.38 (95% CI: 0.22, 0.64; *p*-trend < 0.01) comparing the two extreme tertiles of the updated LDL-C concentrations. Conclusions: In this prospective study, high LDL-C concentrations, when measured closest to RA diagnosis or the end of follow-up, were associated with a low risk of RA. These findings highlight the changes in LDL-C prior to RA diagnosis, and the importance of including lipid analyses into studies of the pathogenesis of RA.

## 1. Introduction

Patients with rheumatoid arthritis (RA) have a higher risk of cardiovascular disease (CVD) than the general population [1,2]. Accordingly, a robust interest in the impact of RA, and RA treatment, on lipid levels—especially low-density lipoprotein cholesterol (LDL-C)—and CVD has developed [3,4]. In contrast, there is relatively less focus, and less clarity, on lipid levels prior to RA diagnosis, or on the potential implications for RA risk. In a study based on a subpopulation of the National Health and Nutrition Examination Surveys, the mean LDL-C and total cholesterol concentrations were lower in RA patients, relative to participants without RA [5]. This may suggest that low LDL-C concentration could be a risk factor for RA. In contrast, a nested case–control study found that women, but not men, with high total cholesterol had an elevated risk of RA compared to those with low–normal total cholesterol [6].

Another explanation is that LDL-C and total cholesterol concentrations decreased significantly prior to RA diagnosis [7]. Previous studies have indicated that metabolites involved in fatty acid metabolism (e.g., acylcarnitine) in the plasma are lower in RA patients prior to diagnosis compared to healthy controls [8,9]. It has been suggested that these changes in fatty acid metabolites may be due to increased beta oxidation, secondary to the inflammation-associated increased energy demands of preclinical RA [8,9]. Increased inflammation has also been proposed to explain the lower LDL-C observed in preclinical and recently diagnosed RA patients [7,10]. While inflammation is known to precede RA diagnosis by several years [11], changes in LDL-C prior to diagnosis, along with the association between LDL-C concentration and RA risk, remain unclear. It is especially important to understand the trajectory of LDL-C prior to RA diagnosis, in order to elucidate whether LDL-C changes in response to the underlying disease processes of RA, or whether LDL-C changes independently and then influences the development of RA.

Thus, in this study we tested whether LDL-C concentrations were associated with the future risk of RA among approximately 100,000 Chinese adults with repeated measures of LDL-C. The associations between other lipids (i.e., high-density lipoprotein cholesterol (HDL-C), triglycerides, or total cholesterol) and RA risk were also explored.

## 2. Methods

### 2.1. Study Population

Participants in the Kailuan Study, an ongoing prospective cohort that began recruitment in 2006, were included in this study. At baseline in 2006, a total of 101,510 participants (between 18 and 98 years old; 81,110 men and 20,400 women) were enrolled. The baseline (2006) study visit and survey were completed between June 2006 and October 2007 by each participant. The survey collected information on medical conditions, medications, demographics, and lifestyle factors, and was re-administered biennially [12,13,14]. Similarly, at baseline and every two years, participants participated in study visits that included laboratory tests and physical examinations by trained nursing staff. At these visits, any changes to participant health were documented on an annual basis. Inclusion criteria for this analysis included complete LDL-C data at baseline, no reported lipid-lowering medication use at baseline or during the follow-up (to remove the potential impact of lipid-lowering medication use on the LDL–RA relationship), and no diagnosis of RA at baseline.

### 2.2. Standard Protocol Approvals, Registrations, and Patient Consent

The Ethics Committee of the Kailuan Medical Group, Kailuan Company, approved this study protocol, and all participants provided their written informed consent. The registration number is 2006-5.

### 2.3. Assessment of Lipid Profile

At each visit, from baseline to 2016, blood samples were collected in the morning, after an overnight fast. In the central laboratory of the Kailuan General Hospital, the enzymatic colorimetric method (Mind Bioengineering Co Ltd., Shanghai, China) was used to measure LDL-C, HDL-C, triglycerides, and total cholesterol. For each measurement, the inter-assay coefficient of variation was <10% with the use of an autoanalyzer (Hitachi 747, Hitachi, Tokyo, Japan) [15].

### 2.4. Assessment of Rheumatoid Arthritis Diagnoses

To confirm that each diagnosis met the 2010 American College of Rheumatology/European League Against Rheumatism (ACR/EULAR) criteria for RA, the medical records of each possible case were reviewed by a team of rheumatologists [14]. The medical records of the participants in the Kailuan Study are held within the Municipal Social Insurance Institution in China, and potential cases of RA were found by querying this institution. Once a potential RA case was identified among the participants of the Kailuan Study, the medical records of these participants were reviewed by three rheumatologists. Cases that met the classification criteria of RA set out by the ACR/EULAR were included as RA cases in this study [16].

### 2.5. Assessment of Covariates

From the same blood draws previously described for lipid quantification, high-sensitivity C-reactive protein (hs-CRP) and blood glucose were also quantified. An autoanalyzer (Hitachi 747; Hitachi, Tokyo, Japan) in the central laboratory of Kailuan General Hospital was used for these measurements.

During the baseline visit, and biennially, the anthropometrics of each participant (e.g., blood pressure, height, weight) were measured by trained nurses [13]. From the height and weight measurements, we calculated body mass index (BMI). Furthermore, during each visit, participants completed questionnaires that recorded self-reported data on sex, age, birthplace, smoking, alcohol consumption, and medical history (e.g., cardiovascular diseases, or current medications, such as antihypertensive and lipid-lowering agents) [17].

### 2.6. Statistical Analysis

In this study, we used SAS, version 9.3 (SAS Institute, Inc., Cary, NC, USA), for our analyses, and a two-sided *p* < 0.05 was considered to be statistically significant.

We used Cox proportional hazards models to investigate the association between tertiles of baseline LDL-C and RA risk, after adjusting for potential confounders, including age, sex, BMI (<23 kg/m^2^, 23–27.5 kg/m^2^, >27.5 kg/m^2^), HDL-C (quartiles), triglycerides (tertiles), diabetes (non-diabetic: blood glucose < 5.6 mmol/L; pre-diabetic: blood glucose 5.6–6.9 mmol/L; diabetic: blood glucose: >6.9 mmol/L, or self-reported diabetes or antidiabetic medication use), hypertension (no hypertension: systolic blood pressure (SBP) < 120 mmHg and/or diastolic blood pressure (DBP) < 80 mmHg; pre-hypertension: SBP 120–139 mmHg and/or DBP 80–89 mmHg; hypertension: SBP > 140 mmHg, DBP > 90, or self-reported hypertension or antihypertensive medication use), alcohol consumption (never or past; light to moderate (women: 0–1.0 servings/day; men: 0–2.0 servings/day); and heavy (women: >1.0 serving/day; men: >2 servings/day)), and smoking (never, past, or current). We used the lowest LDL-C tertile (<2.02 mmol/L) as the reference group. The person-time of each participant was calculated using the time from the baseline visit to the end of follow-up on 31 December 2016, death, or time of diagnosis of RA—whichever came first. The significance of trends was tested by assigning the median concentration of LDL-C in each of the LDL-C tertiles. The proportional hazards assumption was satisfied for the Cox models.

Previous studies have indicated that prior to RA diagnosis, LDL-C may change over time [7,10]. To explore this, we repeated the primary Cox proportional hazard analysis, this time using the updated measurements. Time-varying covariates (e.g., hs-CRP) were used in the analyses.

To examine how LDL-C changes in RA cases before and after diagnosis, we used a residual method previously described by Song et al. [18]. First, using only participants who did not develop RA, we fitted a linear regression model of LDL-C on years to end of follow-up, age at baseline, sex, BMI, HDL-C, and TG. We extracted the beta coefficients from this regression and used them to calculate the expected LDL-C for participants who did develop RA, assuming that this ‘expected LDL-C’ represented the LDL-C that the participants would have had, had they not developed RA. Then, for each participant who developed RA, we calculated the difference between the ‘expected LDL-C’ and actual LDL-C, representing the ‘residuals’ of LDL-C. Finally, we arranged these residuals of LDL-C according to time to diagnosis. If the LDL-C was measured within six months before or after RA diagnosis, this was considered to be measured in the year of diagnosis. We calculated the mean and 95% confidence interval for the residual LDL-C in each time interval (e.g., 0.5–2 years prior to diagnosis, 2–4 years prior to diagnosis) using a mixed-model approach, accounting for repeated measures on each participant.

We also investigated whether sex (categorical: men vs. women), BMI (categorical: <23 kg/m^2^, 23–27.5 kg/m^2^, or >27.5 kg/m^2^), current smoking (categorical: yes vs. no), hs-CRP (categorical: <1 mg/L, 1–3 mg/L, or >3 mg/L), HDL-C (categorical: quartiles), triglycerides (categorical: tertiles), or total cholesterol (categorical: tertiles) affected the association between LDL-C and RA risk using the likelihood ratio test, adjusting for the previously mentioned covariates.

## 3. Results

The present analysis included 97,411 participants (19,566 women and 77,845 men; mean age = 51.8 years). We identified 97 incident cases of RA in the 10 years of follow-up. Participants in the highest LDL-C tertile (≥2.65 mmol/L) were more likely to be men, current smokers, have higher alcohol consumption, and have hypertension (Table 1). In contrast, participants in the highest HDL-C quartile (≥1.77 mmol/L) were more likely to be women, have lower hs-CRP concentrations, and be normoglycemic (Appendix A). The median time to diagnosis of RA was 5.84 years.

The multivariate-adjusted HR of RA was 0.85 (95% CI: 0.53, 1.37) for participants with an LDL-C concentration in the middle tertile (2.02–2.65 mmol/L), and 0.64 (95% CI: 0.38, 1.09) for participants with an LDL-C concentration in the highest tertile (≥2.65 mmol/L), compared to those with LDL-C concentrations in the lowest tertile (<2.02 mmol/L) (*p*-trend = 0.10) (Table 2). Further adjusting this model for hs-CRP generated similar results (Table 2).

Using updated measured LDL-C concentrations as the exposure, the multivariate- and hs-CRP-adjusted HR of RA was 0.56 (95% CI: 0.35, 0.90) for participants with an LDL-C concentration in the middle tertile, and 0.38 (95% CI: 0.22, 0.64) for participants with an LDL-C concentration in the highest tertile, compared to those with LDL-C concentrations in the lowest tertile (*p*-trend < 0.01) (Table 2). Removing hs-CRP from the updated measurements model produced similar results (Table 2).

The residual analysis indicated that in this population, in the 2.5 years prior to RA diagnosis, the mean LDL-C concentration in those who developed RA was less than the expected LDL-C concentration, and the largest difference in LDL-C occurred in the approximately two years prior to RA diagnosis (Figure 1). This residual analysis further indicated that LDL-C concentrations did not continue decreasing relative to the expected LDL-C concentrations after RA diagnosis.

We did not observe a significant association between HDL-C quartiles, triglycerides, or total cholesterol and the HR of RA (Appendix A). There was also no significant interaction between LDL-C tertiles and sex, BMI, smoking, hs-CRP, HDL-C, triglycerides, or total cholesterol in relation to RA risk (*p*-interaction > 0.05 for all).

## 4. Discussion

In this large prospective cohort, we found a significant association between LDL-C and RA risk when using the updated measured values and adjusting for age, sex, smoking status, and other potential RA risk factors. There was no significant association between baseline LDL-C and RA risk, and the results were similarly nonsignificant after adjusting for hs-CRP and including participants who took lipid-lowering medications. Furthermore, we did not find a significant association between other lipids (i.e., HDL-C, triglycerides, or total cholesterol) and the risk of RA.

When we used the updated measures, we found that low LDL-C was significantly associated with high subsequent risk of RA. We also found that HDL-C, triglycerides, and total cholesterol were not significantly associated with RA risk at baseline or when using the updated measures. These results add to a mixed body of literature, which has found inconsistent results regarding lipids prior to RA diagnosis. One case–control study found that LDL-C and total cholesterol concentrations decreased significantly in the five years prior to RA diagnosis, compared to patients who did not develop RA [7]. Meanwhile, another case–control study of serial blood bank samples found that total cholesterol and triglycerides were higher, while HDL-C was lower, in people who went on to develop RA compared to those who did not [19]. The median timing of the first blood sample was 7.5 years prior to diagnosis, and the authors found that the difference in lipid levels between those who developed RA and those who did not varied over time [19]. Additionally, a nested case–control study found that women, but not men, with high total cholesterol had an elevated risk of RA compared to those with low–normal total cholesterol [6]; in this case–control study, the median time to diagnosis was 12 years [6]. The mixed results in the literature may be due to the varying times between lipid assessment and RA diagnosis. This theory is supported by our residual analysis, in which we found that LDL-C concentrations in those who developed RA were lower than expected in the few years before diagnosis, but not 5 or 10 years prior to diagnosis. In these RA patients, LDL-C was approximately 0.2 mmol/L lower prior to diagnosis. For this change in LDL-C to be identified clinically, periodic—perhaps annual—cholesterol testing would be required, which may not be feasible outside of cohort studies.

Blood lipids, especially LDL-C, could be affected by the inflammation and autoimmunity that are part of the pathogenesis of RA [7,10]. In a comparison, during the years 1999–2002 and 2007–2010, of RA patients in the Boston area with participants in the National Health and Nutrition Examination Surveys (NHANES), none of whom were taking lipid-lowering medications at the time of lipid quantifications, the mean LDL-C and total cholesterol concentrations were lower in RA patients (mean LDL-C (mg/dL): 111 versus 122 in 1999–2002, 105 versus 118 in 2007–2010, in RA patients and NHANES participants, respectively) [5]. This observation has been replicated in a clinical trial that investigated the effects of atorvastatin and simvastatin on lipid levels and cardiovascular events in participants, with and without RA, who had a previous myocardial infarction [20]. The study found that those with RA had lower baseline LDL-C and total cholesterol than participants without RA, though both groups responded similarly to statin treatment [20].

Furthermore, in clinical trials of various therapies for RA (e.g., methotrexate, tocilizumab, tofacitinib) that decrease systemic inflammation, increases in total cholesterol and LDL-C have been observed after treatment [21,22,23]. Stable isotope studies of lipid metabolism in RA patients have indicated that the lower baseline LDL-C levels in RA patients compared to non-RA patients may be due to increased cholesterol catabolism, or removal from the bloodstream [21,24]. Furthermore, these studies demonstrate that treatment-associated increases in LDL-C may be due to decreased catabolism of cholesterol [21,24]. These results are supported by metabolomic studies that have found that metabolites involved in fatty acid metabolism—such as acylcarnitine—in the plasma are lower in RA patients prior to diagnosis compared to healthy controls [8,9]. These changes in fatty acid metabolites could be due to increased beta oxidation, secondary to the inflammation-associated increased energy demands of preclinical RA [8,9]. As it is possible that the disease processes prior to RA diagnosis are unique compared to established RA, further study of lipid levels prior to RA diagnosis is required.

It is not currently clear whether the elevated CVD risk observed in RA patients is independent of LDL-C, and instead caused by other factors (e.g., elevated systemic inflammation). As LDL-C is a key risk factor for CVD in the general population, it is important for future studies to investigate the relationship between LDL-C and CVD risk in RA. The strengths of this study include the large cohort size and the fact that the RA cases in this study were all rheumatologist-confirmed. The prevalence of RA in Asia is lower than in Europe and North America, and the prevalence in our population is consistent with the previously reported prevalence of RA in China [25]. Additionally, the repeated quantification of blood lipids and the relatively long follow-up allowed for multiple analytical approaches. This majority-male and -Chinese cohort is also unique in the RA prospective cohort literature, which relies mainly on majority-female European-descendent populations.

This study also has several limitations. The generalizability of the findings is limited because the participants in this cohort are not representative of diverse races or socioeconomic classes. While we excluded participants who were taking lipid-lowering medications at baseline and during follow-up, we cannot exclude the possibility of under-reporting medication use. We do not expect that this could substantially impact our results, as the prevalence of lipid-lowering medication use in China is low [26,27]. It was also not possible to account for other important RA risk factors not measured in this study, including family history and exposure to silica dust [28]. Furthermore, while cigarette smoking has been reported to be associated with an elevated risk of RA, [28] we did not observe a significant effect of smoking on RA risk in this cohort. This may be due to the low incidence of RA in this Chinese population (lack of statistical power), or to the limitation that our measure of smoking did not account for the duration or amount of cigarettes smoked on average. Because of the small sample size of incident RA cases, we cannot perform detailed subgroup analyses. However, the small number of cases is consistent with previous reports of the prevalence of RA in China [25]. Finally, it would be interesting to investigate the impact of dietary intake and exercise on the relationship between LDL-C and RA risk; however, diet and exercise data were not available for this analysis.

## 5. Conclusions

In conclusion, our study demonstrates that low LDL-C may be associated with elevated RA risk in the near future. These findings highlight the potential changes in LDL-C concentration that may occur before an RA diagnosis, especially in the years immediately prior to the diagnosis. The proximity of these lipid level changes to the diagnosis of RA may indicate that changes in LDL-C occur secondary to the underlying RA disease processes, but further research is needed in order to elucidate the relationships between autoimmunity, inflammation, and LDL-C in RA.

## Figures and Tables

**Figure 1 nutrients-14-01240-f001:**
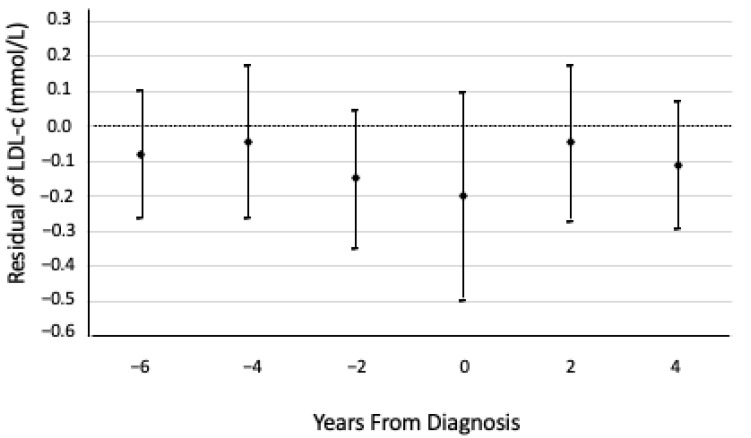
Means and 95% confidence intervals of low-density lipoprotein cholesterol (LDL-C) residuals relative to the expected LDL-C concentrations had those cases not developed rheumatoid arthritis (RA), over time, from the date of RA diagnosis. Residuals are based on a linear regression model—using only participants who did not develop RA—of LDL-C on years to end of follow-up, adjusted for age at baseline, sex, BMI, HDL-C, and TG, and accounting for repeated measures on each participant. The expected LDL-C concentrations from timepoints −6 to 4 were 2.39, 2.50, 2.62, 2.72, 2.83, and 2.95 mmol/L, respectively.

**Table 1 nutrients-14-01240-t001:** Baseline characteristics in 2006 by low-density lipoprotein cholesterol concentration, among 97,687 Kailuan Study participants without rheumatoid arthritis, adjusted for age and sex ^a^.

	Low-Density Lipoprotein Cholesterol		
	<2.02 mmol/L	2.02–2.65 mmol/L	≥2.65 mmol/L
N	32,727	32,251	32,433
Women, %	25.0	18.3	16.8
Age, year	52.2 ± 0.07	49.3 ± 0.08	50.4 ± 0.08
Alcohol intake, grams/day	4.32 ± 0.14	4.86 ± 0.14	7.27 ± 0.14
Smoking status, %			
Never	65.9	60.0	53.2
Past	5.57	5.88	6.38
Current	28.5	34.1	40.4
CRP ^b^, mg/L	3.09 ± 0.04	2.03 ± 0.04	2.13 ± 0.04
BMI ^c^, kg/m^2^	24.5 ± 0.02	25.0 ± 0.02	25.3 ± 0.02
Diabetes status, %			
Normoglycemic	77.6	69.2	64.0
Pre-diabetic	14.2	22.1	25.6
Diabetic	8.18	8.75	10.4
Hypertension status, %			
Normotensive	22.7	20.3	16.4
Pre-hypertensive	48.6	49.2	48.4
Hypertensive	28.8	30.6	35.3
LDL-C ^d^, mmol/L	-	-	-
HDL-C ^e^, mmol/L	1.57 ± 0.00	1.58 ± 0.00	1.56 ± 0.00
Triglycerides, mmol/L	1.63 ± 0.01	1.56 ± 0.01	1.60 ± 0.01

^a^ Continuous variables are presented as the mean ± standard error, adjusted for sex and age. Low-density lipoprotein cholesterol concentrations are split into tertiles. Age is only adjusted for sex. ^b^ C-reactive protein. ^c^ Body mass index. ^d^ Low-density lipoprotein cholesterol (LDL-C). ^e^ High-density lipoprotein cholesterol (HDL-C).

**Table 2 nutrients-14-01240-t002:** Adjusted hazard ratios and 95% confidence intervals for risk of rheumatoid arthritis by low-density lipoprotein cholesterol concentration.

	Tertile of LDL-C Concentrations			*p*-Trend
**Baseline LDL-C**	**T1 (<2.02 mmol/L)**	**T2 (2.02–2.65 mmol/L)**	**T3 (≥2.65 mmol/L)**	
# of case/population	43/32,727	31/32,251	23/32,433	
Incidence rate (/10,000 person-years)	1.64	1.19	0.89	
Sex- and age-adjusted hazard ratio	1.00 (Ref.)	0.87 (0.55, 1.39)	0.64 (0.39, 1.07)	0.09
Multivariate-adjusted ^a^	1.00 (Ref.)	0.85 (0.53, 1.37)	0.64 (0.38, 1.09)	0.10
Multivariate-adjusted and hs-CRP-adjusted ^b^	1.00 (Ref.)	0.87 (0.54, 1.39)	0.66 (0.39, 1.11)	0.12
**Updated LDL-C**	**T1 (<2.42 mmol/L)**	**T2 (2.43–3.11 mmol/L)**	**T3 (≥3.12 mmol/L)**	
Multivariate-adjusted ^c^	1.00 (Ref.)	0.56 (0.35, 0.90)	0.38 (0.22, 0.64)	< 0.01
Multivariate-adjusted and hs-CRP-adjusted ^d^	1.00 (Ref.)	0.58 (0.36, 0.94)	0.39 (0.23, 0.67)	< 0.01

^a^ Adjusted for sex, age, body mass index (<18.5 kg/m^2^, 18.5–23 kg/m^2^, 23–27.5 kg/m^2^, or >27.5 kg/m^2^), anti-hypercholesterolemia drug use (yes vs. no), high-density lipoprotein cholesterol (HDL-C) (quartiles), low-density lipoprotein cholesterol (LDL-C) (tertiles), triglycerides (tertiles), alcohol consumption (never or past, light to moderate (women: 0–1.0 servings/day; men: 0–2.0 servings/day), or heavy (women: >1.0 serving/day; men: >2 servings/day)), smoking (never, past, or current), diabetes (non-diabetic, pre-diabetic, or diabetic), and hypertension (no hypertension, pre-hypertension, or hypertension). ^b^ Adjusted for the above covariates and further adjusted for high-sensitivity C-reactive protein (hs-CRP) (<1 mg/L, 1–3 mg/L, or >3 mg/L). ^c^ Adjusted for the above covariates, using the updated measurements before the end of follow up, death, or rheumatoid arthritis diagnosis. ^d^ Adjusted for the above covariates, using the updated measurements before the end of follow up, death, or rheumatoid arthritis diagnosis, and further adjusted for high-sensitivity C-reactive protein (hs-CRP) (<1 mg/L, 1–3 mg/L, or >3 mg/L).

## Data Availability

Data described in the manuscript, code book, and analytic code will be made available upon request by contacting Drs. Cui and Gao.

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
