# Peer review of "Low-Density Lipoprotein Cholesterol and the Risk of Rheumatoid Arthritis: A Prospective Study in a Chinese Cohort"

_nutrients, 2022, doi:10.3390/nu14061240_

Round 1

Reviewer 1 Report

In this manuscript the Authors tested the hypothesis that low LDL is a RA risk factor. The topic is interesting and in general the message is easy to understand. The title is quite clear as well as the aim; methods are well described and conclusions are consistent with results.

COMMENTS

- As the Authors underlined in the main text, this Chinese cohort can be quite different from a Caucasian one. So I suggest to add this point to the title (e.g. “… a prospective Chinese study”).

- Even if the aim is clear, the statement should be sharper: e.g. “… LDL concentrations increase the RA onset risk in Chinese adults”. Moreover the number of enrolled patients is not appropriate in this section

- L73-77 the baseline cohort characteristics must be moved in results. In this section should be reported the inclusion criteria (e.g. not RA diagnosis at baseline, no lipid-lowering medication etc)

- L96 included —> diagnosed as RA?

- L105 it is redundant how to calculate the BMI 

- L 144-115 it is not clear if BMI is a categorical or continues variable

- L116-117 provide the definition of both the conditions (pre hypertension, pre diabetes)

- I find quite unusual that smoke did not have any impact on RA onset. Please, deeply discuss this point. Moreover other common risk factor of RA were not taken into account (e.g. familiar history etc). Even if this issue could represent a huge bias, this study type do not allow to explore some points. However the Author must consider it as a limit. 

Author Response

- As the Authors underlined in the main text, this Chinese cohort can be quite different from a Caucasian one. So I suggest to add this point to the title (e.g. “… a prospective Chinese study”).

Thank you for this suggestion, we have edited the title accordingly. It now reads “Low density lipoprotein cholesterol and risk of rheumatoid arthritis: a prospective study in a Chinese cohort”.

- Even if the aim is clear, the statement should be sharper: e.g. “… LDL concentrations increase the RA onset risk in Chinese adults”. Moreover the number of enrolled patients is not appropriate in this section

Thank you for you suggestion. We have clarified the population like you have suggested, this sentence now reads “To investigate whether low density lipoprotein cholesterol (LDL-C) concentration was associated with risk of rheumatoid arthritis (RA) in Chinese adults”. We did not use “increase the .. risk” because this is an observational study and we tried to avovid the use of causal langague.

We typically report the number of participants in the cohort in the abstract as it would help readers to interpret the observed results, but can remove it.

- L73-77 the baseline cohort characteristics must be moved in results. In this section should be reported the inclusion criteria (e.g. not RA diagnosis at baseline, no lipid-lowering medication etc)

Thank you for your suggestion. We have moved the following sentence to the beginning of the results section as you suggested:  “The current analysis included the 97,411 participants (19,566 women and 77,845 men; mean age=51.8 y)”.

We have also added the following sentence to the methods section, “Inclusion criteria for this analysis included complete LDL-C data at baseline, no reported lipid-lowering medication use at baseline and during the follow-up (to remove the potential impact of lipid-lowering medication use on the LDL-RA relationship), and no diagnosis of RA at baseline”.

- L96 included —> diagnosed as RA?

Thank you for mentioning this, we have clarified that these participants were included as RA cases by clarifying as follows, “Cases that met the classification criteria of RA set out by ACR/EULAR were included as RA cases in this study”.

- L105 it is redundant how to calculate the BMI 

Thank you for this comment, we have removed the calculation. The sentence now reads, “From the height and weight measurements, we calculated  body mass index (BMI)”.

- L 144-115 it is not clear if BMI is a categorical or continues variable

Thank you for pointing this out, we have clarified the variables in this sentence. It now reads, “We also investigated whether sex (categorical, men vs women), BMI (categorical, < 23 kg/m2, 23-27.5 kg/m2, > 27.5 kg/m2), current smoking (categorical, yes vs no), hs-CRP (categorical, <1 mg/L, 1-3 mg/L, > 3 mg/L), HDL-c (categorical, quartiles), triglycerides (categorical, tertiles), or total cholesterol (categorical, tertiles) affected the association between LDL-C and RA risk using the likelihood ratio test, adjusting for the previously mentioned covariates”.

- L116-117 provide the definition of both the conditions (pre hypertension, pre diabetes)

Thank you for this comment, we have revised this section as follows, “We used Cox proportional hazards models to investigate the association between tertiles of baseline LDL-C and RA risk, after adjusting for potential confounders, including age, sex, BMI (< 23 kg/m2, 23-27.5 kg/m2, > 27.5 kg/m2), HDL-c (quartiles), triglycerides (tertiles), diabetes (non-diabetic: blood gluose < 5.6 mmol/L, pre-diabetic: blood glucose 5.6-6.9 mmol/L, diabetic: glood glucose: > 6.9 mmol/L or self-reported diabetes or antidiabetic medication use), hypertension (no hypertension: systolic blood pressure (SBP) < 120 mmHg and/or diastolic blood pressure (DBP) < 80 mmHg, pre-hypertension: SBP 120-139 mmHg and/or DBP 80-89 mmHg, hypertension: SBP > 140 mmHg, DBP > 90 or self-reported hypertension or antihypertensive medication use), alcohol consumption never or past, light to moderate (women: 0–1.0 servings/d; men: 0–2.0 servings/d), and heavy (women: >1.0 serving/d; men: >2 servings/d), and smoking (never, past, current)”.

- I find quite unusual that smoke did not have any impact on RA onset. Please, deeply discuss this point. Moreover other common risk factor of RA were not taken into account (e.g. familiar history etc). Even if this issue could represent a huge bias, this study type do not allow to explore some points. However the Author must consider it as a limit. 

This is a good point, thank you for bringing this to our attention. We have added the following to the discussion, “It was also not possible to account for other important RA risk factors not measured in this study, including family history and exposure to silica dust. Further, while cigarette smoking has been reported to be associated with an elevated risk of RA, we did not observe a significant effect of smoking on RA risk in this cohort. This may be due to the low incidence of RA in this Chinese population (lack of statistical power) or the limitation that our measure of smoking did not account for duration or amount of cigarettes smoked on average”.

Reviewer 2 Report

The authors investigated that the relationship between the onset of rheumatoid arthritis (RA) and low-density lipoprotein cholesterol (LDL-C) levels in the blood and found that LDL-C was decreased before the diagnosis of RA in the epidemiological study in China. In my opinion, such continuous and large-scale epidemiological studies are important for understanding the disease. Please consider addressing the following comments.

  1. In Figure 1, the expected concentration of LDL-C used as a baseline should be given in the specific values. A maximam decrease of about 0.2 mmol/L was observed before the diagnosis of RA, which seem to be a small change compared with the case of other diseases such as CVD. The authors should discuss whether the small change is realistically possible to diagnose.
  2. While 80% of the population in this study was male, RA is a common disease in women. With regard to the relationship between lowering LDL-C and the onset of RA reported in this study, is there any effect of gender differences, such as a stronger correlation in women?
  3. The relationship between lifestyle, including diet and exercise, and LDL-C levels and the risk of onset RA should also be considered.
  4. As described in introduction section, RA is a risk factor for the onset and progression of CVD. On the other hand, LDL-C is also a risk factor for CVD, but is decreased before RA diagnosis, which seems to be a contradiction. What do the authors discuss this?

Author Response

-In Figure 1, the expected concentration of LDL-C used as a baseline should be given in the specific values. A maximam decrease of about 0.2 mmol/L was observed before the diagnosis of RA, which seem to be a small change compared with the case of other diseases such as CVD. The authors should discuss whether the small change is realistically possible to diagnose.

This is a good point, thank you for bringing it to our attention. We have added the following text to the discussion, “In these RA patients, LDL-C was approximately 0.2 mmol/L lower prior to diagnosis. For this change in LDL-C to be identified clinically, periodic, perhaps annual, cholesterol testing would be required, which may not be feasible outside of cohort studies”. On Figure 1, we have added the expected concentrations to the caption, so as not to obscure the figure. This addition reads, “The expected LDL-C concentrations from timepoint -6 to 4 were 2.39, 2.50, 2.62, 2.72, 2.83, 2.95 mmol/L respectively”.

-While 80% of the population in this study was male, RA is a common disease in women. With regard to the relationship between lowering LDL-C and the onset of RA reported in this study, is there any effect of gender differences, such as a stronger correlation in women?

This is also a good point. We tested the interaction between sex and LDL-C on RA risk, and found no significant effect of sex on the association between LDL-C and RA risk. We have the following text in the results section, “There was also no significant interaction between LDL-C tertiles and sex, BMI, smoking, hs-CRP, HDL-c, triglycerides, or total cholesterol in relation to RA risk (P interaction > 0.05 for all)”.

-The relationship between lifestyle, including diet and exercise, and LDL-C levels and the risk of onset RA should also be considered.

This is true, it would be beneficial to examine the impact of diet and exercise on LDL-C and RA risk. As we did not have exercise or diet measures, we have added the following to the discussion section, “Finally, It would be interesting to investigate the impact of dietary intake and exercise on the relationship between LDL-C and RA risk, however diet and exercise data were not available for this analysis”.

-As described in introduction section, RA is a risk factor for the onset and progression of CVD. On the other hand, LDL-C is also a risk factor for CVD, but is decreased before RA diagnosis, which seems to be a contradiction. What do the authors discuss this?

Thank you for bringing this up, we have added the following to the discussion, “It is not currently clear whether the elevated CVD risk observed in RA patients is independent of LDL-C, and instead caused by other factors (e.g. elevated systemic inflammation). As LDL-C is a key risk factor for CVD in the general population, it is important for future studies to investigate the relationship between LDL-C and CVD risk in RA”.

Round 2

Reviewer 1 Report

All comments were addressed

Reviewer 2 Report

The revised manuscript provides better content to understand this topic.